# Harnessing Chitin from Edible Insects for Livestock Nutrition

**DOI:** 10.3390/insects16080799

**Published:** 2025-08-01

**Authors:** Linda Abenaim, Barbara Conti

**Affiliations:** Department of Agriculture, Food and Environment, Via del Borghetto 80, 56124 Pisa, Italy; barbara.conti@unipi.it

**Keywords:** feed, polysaccharide, digestibility, edible insects, prebiotic effect, beneficial effect, livestock animals

## Abstract

The utilisation of edible insects as sustainable animal feed is a recent development that shows great promise. These insects offer a rich profile of nutrients, including proteins, lipids, vitamins, and chitin. Chitin remains relatively underexplored despite its potential impact on animal health and performance. This natural polysaccharide, which is found in abundance in insect exoskeletons, plays a dual role in animal nutrition. On the one hand, it poses digestibility challenges. Chitin may contribute to improved gut health, immune modulation, and reductions in cholesterol and pathogenic bacteria. The present review analyses the current state of research regarding the inclusion of insect-derived chitin in the diets of authorised insect species for animal nutrition.

## 1. Introduction

Over the past ten years, the use of edible insects as a sustainable and innovative resource for animal nutrition has gained significant attention due to their potential to address global challenges such as population growth, increasing food demand, the need for alternative protein sources, and climate change [1]. In this regard, edible insects align with the Sustainable Development Goals by 2030 Agenda [2] by offering an eco-friendly alternative to conventional protein sources, as some of them can transform organic waste into high-value protein products. Their consumption is due principally to their excellent nutritional profile, making them suitable for both human and animal diets. In parts of the world such as Asia, Africa, and Latin America, where entomophagy is a traditional practice, insects are often consumed whole and prepared using various techniques such as frying, boiling, and drying. In contrast, in European countries, where their consumption is legally regulated [3], insects are primarily processed into protein flours, to enrich foods with low-protein content and enhance their nutritional value [1]. However, the most significant and widespread use of edible insects occurs as a feed ingredient, particularly in aquaculture, livestock (poultry and pigs), and the pet industry (including dogs, cats, fish, and reptiles), where they are commonly incorporated in the form of insect meal [4]. Edible insects are rich in proteins, amino acids, fatty acids, minerals, vitamins, chitin, and other bioactive compounds for human and animal nutrition [1,5,6,7]. Proteins constitute a major component, ranging from 30 to 65% of their total dry matter, with the amino acids’ composition varying between 46 and 96%, characterised by a high level of essential amino acids such as phenylalanine, tyrosine, tryptophane, lysine, and threonine [8,9]. Lipids represent the second major component, ranging from 13 to 33%, with significant levels of polyunsaturated fatty acids (PUFAs) and omega-3 fatty acids [8,9]. Additionally, many insect species provide considerable amounts of micronutrients such as calcium, zinc, and iron [10,11]. Beyond their nutritional value, edible insects also contain bioactive compounds, among which chitin stands out and represents at least 10–15% of all dry matter [12,13]. Among the various components derived from edible insects, chitin is the least studied component despite its unique properties and potential benefits. Chitin is a natural polysaccharide found in the procuticle of the insect exoskeleton, the uppermost layer of the epidermis [14]. Chitin presents both negative and positive aspects in animal nutrition. As is known, its digestibility varies among animal species due to differences in chitinase production, which can limit its bioavailability [15,16,17,18,19,20]. However, studies suggest that chitin may offer several benefits, including modulation of gut microbiota, improvement of the immune response, reduction in cholesterol and reduction in some pathogens of certain animal species [13,17,18,20,21]. Given these contrasting aspects, this review aims to explore the role of chitin derived from edible insects in animal nutrition, focusing on both its negative and positive effects across different animal species.

## 2. Structure and Role of Insect-Derived Chitin

The exoskeleton of an insect is a complex and multilayered structure that provides protection, support, and a surface for muscle attachment. It is made primarily of chitin combined with proteins and sometimes minerals. In synthesis, the main layers of the insect exoskeleton are as follows: a. cuticle, b. epidermis, and c. basement membrane (Figure 1).

The cuticle is formed, from outside to inside, by the epicuticle, the outermost, thin layer. It waterproofs the insect and prevents dissection, as it is mainly composed of waxes. It may also contain pigments. The layer beneath the epicuticle is the exocuticle. It is the sclerotised part of the cuticle, often pigmented. It contains a few chitin fibres, which are cross-linked with proteins, providing strength and rigidity to the exoskeleton. The deeper layer of the cuticle is the endocuticle, which is thicker and softer than the exocuticle, as it is composed of chitin and non-sclerotised proteins. It provides flexibility and helps absorb mechanical stress. Chitin microfibres are arranged in a helical or layered pattern to increase strength. The exocuticle and endocuticle constitute the so-called procuticle.Below the cuticle is the epidermis, a living, glandular cell layer that secretes all the layers above it. It is responsible for producing chitin, proteins, and wax.The basement membrane is a thin layer beneath the epidermis that separates the exoskeleton from the rest of the insect’s body tissues [22].

Chitin (C_8_H_13_O_5_N)_n_ is the second most important polymer after cellulose made by long-chain of N-acetyl-β-D-glucosamine joined by 1,4 covalent bonds (Figure 2). It was discovered in mushrooms by Henri Braconnot in 1811 who named it “fungine” and subsequently was named “chitin” by Auguste Odier in 1821 who isolated it from beetle cuticles [23]. It is one of the main structural components of the exoskeleton of arthropods, and therefore also of insects’ cuticles, as well as the structural membranes and spores of fungi and yeasts [14,24,25,26].

Due to its high crystallinity and extensive hydrogen bonding, chitin is highly insoluble in water and most organic solvents; however, it can be partially degraded in highly concentrated acids (e.g., HCl) or certain ionic liquids [14,27,28]. These characteristics contribute to its strength, low density, water resistance, and chemical stability, making it the key structural component in insect exoskeleton cuticle. Kramer et al. [29] revealed that chitin constituted up to 40% of the exuvial dry mass depending on the insect species and their development. In insects, chitin is found principally in the unsclerotised endocuticle of the exoskeleton, and, as a consequence, it can also be found in the other structures of the insect body of ectodermic origin like the fore and hindgut, mesenteron peritrophic matrix, salivary gland, tracheae, and eggshells [14,24,30]. The function of chitin in the insects’ body wall is principally to give stability and elasticity to the structure, while in the peritrophic matrix, it works as a permeability barrier between food and the midgut epithelium protecting the brush border from mechanical abrasion [24,30]. Chitin microfibrils (about 3 nm in diameter) are stabilised by hydrogen bonds between amine and carbonyl groups and, according to various microfibril orientations, chitin can exist in three different crystalline forms: α-, β-, and γ-chitin. The main differences between these forms are the disposition of the polymeric chain, the degree of hydration, and mechanic properties [14,24,28,30]. All three crystalline forms are found in the cuticle of insects. In particular the form α- (with an anti-parallel orientation) is the predominant form in procuticle, while the β- (with a parallel orientation) and γ- forms (with two parallel strands alternated with anti-parallel strands), beside procuticle, are also frequently found in cocoons produced in some species (such as Neuroptera and Thysanoptera) through secretion from the Malpighian tubules [14,24,30,31]. The peritrophic matrix is usually characterised by microfibrils with α- and β- orientations [32].

The formation of chitin in insects is catalysed by chitin synthases, membrane-bound glycosyltransferases that polymerise Uridine-Diphosphate-N-acetylglucosamine (UDP-GlcNac) into chitin chains. The role of chitin synthase is essential for insect growth (for moulting and metamorphosis), development, and reproduction. This enzyme is tightly regulated by two major isoforms: *CHS1* or *CHS-A*, primarily expressed in epidermal cells and tracheal epithelium, is responsible for cuticle and exoskeleton formation, while *CHS2* or *CHS-B*, expressed in gut epithelial cells, is responsible for production of peritrophic matrix-associated chitin [24,30,33,34]. During moulting, the breakdown and structural changes in chitin are further controlled by chitinases, proteases, and chitin deacetylases. These enzymes are collectively referred to as exuvial fluid [14,24,30,33].

## 3. Chitin Content in Edible Insects

Although there are over 2000 edible insect species out of over one million, only a few hundred have been studied for their potential in animal feed applications. While Africa, Asia, and Latin America lead in edible insect production, interest is growing also in Europe and Oceania, particularly following the approval of certain insect species for feed use under two European Regulations [35,36]. The most consumed insects for food and feed are beetles (Coleoptera, 32%), caterpillars (Lepidoptera, 18%), bees, wasps and ants (Hymenoptera, 14%), grasshoppers, locusts and crickets (Orthoptera, 13%), and flies (Diptera, 2%) [1]. Among these, *Hermetia illucens* (Linnaeus, 1758) (Diptera Stratiomyidae), *Musca domestica* (Linnaeus, 1758) (Diptera Muscidae), *Tenebrio molitor* (Linnaeus, 1758) (Coleoptera Tenebrionidae), *Alphitobius diaperinus* (Panzer, 18797) (Coleoptera Tenerbionidae), *Acheta domesticus* (Linnaeus, 1758) (Orthoptera Gryllidae), *Gryllodes sigillatus* (Walker, 1869) (Orthoptera Gryllidae), *Gryllus assimilis* (Frabricius, 1775) (Orthoptera Gryllidae), and *Bombyx mori* (Linnaeus, 1758) (Lepidoptera Bombycidae) are the species allowed in the European Union for animal feed production, in aquaculture, poultry, and swine farming (Figure 3) [14,37,38,39].

These insects are rich in proteins and lipids and contain significant amounts of chitin. The most investigated edible insect species for their chitin content are *H. illucens*, *T. molitor*, *A. domesticus*, and *B. mori* [40,41,42,43]. The variation in chitin content depends on the insect species and the biological stages: *T. molitor* and *B. mori* possess chitin levels of 16 to 20%, *H. illucens* ranges from 10 to 20%, whilst *A. domesticus* displays a reduced chitin content of 4 to 7% [27,28]. Furthermore, the developmental stage has been demonstrated to affect the chitin content. For instance, the chitin concentration in *H. illucens* larvae is approximately 9%, whereas in exuviae it is 24% [44], but it is approximately 22% in *T. molitor* adults and 4% in larvae [45]. Table 1 presents the chitin content of the eight edible insect species authorised for animal feed in the EU at their respective biological stages.

## 4. Importance of Insect-Derived Chitin on Animal Nutrition: Disadvantages and Benefits

The nutritional impact of insect-derived chitin remains a topic of debate, as it exhibits both beneficial and adverse effects on animal nutrition. On one hand, chitin is an insoluble and poorly digestible fibre, which may hinder feed efficiency and nutrient absorption [13,15,18,19,21]; on the other, recent studies highlighted its beneficial role, especially in the prebiotic effect and modulation of intestinal microbiota, promotion of the immunologic system, and anti-inflammatory and antioxidant capacity [13,17,18,21,55]. This paragraph focuses specifically on the eight insect species currently authorised in the European Union for use in animal feed, with particular emphasis on the effects of their chitin content in fish, poultry, and swine nutrition.

### 4.1. Disadvantages of Insect-Derived Chitin in Animal Nutrition

It is evident that chitin, due to its complex structure, can limit its digestibility in several animals, especially the monogastric ones. The major limiting factor is the lack of endogenous enzymatic activity able to break down chitin into N-acetylglucosamine monomer, which is more digestible and can be effectively absorbed and utilised by the animal. Chitinase production varies widely between animals, and its presence and effectiveness are essential for chitin digestion and to enhance the nutritional value of insect meal [20]. Some fish and crustaceans, which naturally consume levels of chitin in their diet, express chitinase activity in their digestive tracts to process insect-derived chitin effectively [20]. However, the ability of fish to digest chitin is still controversial. Although some carnivorous and omnivorous fish species possess this chitinolytic enzyme, high inclusion levels of insect meal may exceed their digestive capability, resulting in incomplete chitin degradation and compromising the nutrient digestibility and assimilation [16,56,57,58,59,60,61,62,63,64]. In this context, insect meal inclusion levels are generally classified as low (0–5%), moderate (5–30%), and high (>30%), based on their proportion in the total feed. Numerous studies have reported that insect meal inclusion level above 30% can negatively affect the apparent digestibility coefficients (ADC) of nutrients, as well as fish growth and overall performance [56,57,59,61,63,64,65,66]. These effects are often associated with excessive chitin content in the insect meal (usually above 1%) which may hinder the digestion and absorption of essential nutrients such as crude proteins (CP), amino acids (AAs), and ether extract (EE, the measure of lipid content), by interacting with digestive enzymes like chitinase, proteinase and lipase [67]. Several studies have demonstrated a dose-dependent reduction in nutrient digestibility as chitin content increases. For instance, in rainbow trout and meagre, increasing the inclusion level of *H. illucens* or *T. molitor* was associated with progressively lower ADC values for proteins, gross energy (GE), and amino acids, particularly when the chitin fraction ranged between 1.5% and 7% [61,62,66]. Similar findings have been observed in species such as gilthead seabream and African catfish hybrids [59,63]. Nevertheless, several exceptions have been observed, which underscore the interaction between chitin content and nutrient bioavailability. For instance, Atlantic salmon and Nile tilapia exhibited notable levels of tolerance to *H. illucens* and *T. molitor* inclusion levels of up to 100% and 43%, respectively, without significant impacts on nutrient digestibility [58,68]. In a similar way, European seabass and juvenile red sea bream exhibited high tolerance to diets comprising up to 45% of *H. illucens* meal [69,70], indicating that certain species may possess more efficient chitin-degrading mechanisms or benefit from gut microbiota adaptations. Among the eight edible insect species currently approved in the EU for aquaculture feed, *H. illucens* and *T. molitor* are the most widely investigated. However, some recent studies that are emerging on the other species, including *A. diaperinus* larvae and *G. assimilis* adults, indicate a marked decrease in ADC at 5 and 7% chitin inclusion in rainbow trout and Nile tilapia [60,64]. In contrast, insect meals derived from *G. sigillatus* adults, *M. domestica* larvae, and *B. mori* chrysalides appear to be better tolerated by various fish species such as rainbow trout, Nile tilapia, common carp, and Juvenile mirror carp, even at relatively high inclusion level [71,72,73,74]. Table 2 provides an overview of current studies evaluating the effects of insect-derived chitin on nutrient digestibility across different fish species.

As in fish, poultry possess endogenous chitinolytic activity, particularly through the secretion of acidic chitinase in the proventriculus, with lower enzymatic expression observed in other segments of the gastrointestinal tract [81,82]. Despite this, the overall digestibility of insect-derived chitin in poultry remains limited, depending on the dietary inclusion level and the specific chitin concentration of the insect species used. Several studies have demonstrated that elevated levels of insect meal inclusion can result in increased intestinal viscosity, consequently leading to diminished nutrient absorption, particularly of proteins and lipids. This, in turn, has been shown to have a deleterious effect on feed efficiency and growth performance [16,18]. For instance, inclusion levels from 15 to 100% of *H. illucens* and *T. molitor* in broiler and laying hens diets have been associated with low ADC of dry matter, proteins, and amino acids [83,84,85,86]. Conversely, lower or moderate inclusion level (generally from 0.2 to 30%) often resulted in neutral or even positive effects, suggesting a potential mechanical stimulation of the gizzards and increase in secretion of digestive fluids [87,88,89,90,91,92,93,94,95,96]. Research on other EU-authorised insect species is still limited in poultry, but preliminary data suggest diverse digestibility outcomes. For instance, *M. domestica* larval meal showed high protein and amino acid digestibility in broilers at moderate inclusion levels [87] while *G. assimilis* nymphs and *B. mori* chrysalides appear to reduce nutrient digestibility at inclusion levels above 10%, especially in quails [84,97]. On the other hand, *B. mori* chrysalids meal included at lower levels (e.g., 4%) has shown no adverse effects in broilers [98]. This variability is clearly outlined in Table 3, which presents a detailed synthesis of current studies evaluating the effect of insect-derived chitin on nutrient digestibility in poultry, across the edible insect species authorised for feed use in EU.

Even swine seems to secrete chitinase enzyme in their gastric tissue [99,100]. Although there are few studies on the effects of insect chitin on digestibility of swine, it is largely dependent on the insect species, the inclusion level in the diet, and the chitin content. Several studies evaluated *H. illucens* larvae meal, showing contrasting outcomes. Phaengphairee et al. [101] observed a high ADC of dry matter, proteins, and ether extract in piglets-fed diets containing either 12% or 100% *H. illucens* larval meal, characterised by a chitin content of approximately 0.52%. Similarly, Biasato et al. [102] reported no adverse effect on the ADC of dry matter, proteins, and ether extract in piglets-fed diets supplemented with 5% and 10% of inclusion levels, with chitin content ranging from 0.27% to 0.53%. In contrast other studies suggest potential digestion limitations. Liu et al. [103] observed a dose-dependent reduction in the ether extract digestibility when piglets were fed with a complete inclusion of *H. illucens* larvae (100%) corresponding to a high chitin content (1.38%). Furthermore, Yu et al. [104] reported a reduction in protein and fat digestibility even at lower inclusion levels of *H. illucens* larvae (1%, 2%, and 4%). For that which concerns *T. molitor*, some authors reported a high ADC of nutrients in weaning and growing pigs-fed diets containing a low larval inclusion level (from 1.5% to 10%) [105,106,107]. In newly weaned piglets, Pereira et al. [108] showed that a moderate inclusion of 30% *T. molitor* larval meal did not negatively affect protein and amino acid digestibility. Limited data are available for other EU-approved insect species. A study by Tan et al. [109] revealed that a very high dietary inclusion (97.67%) of *M. domestica* larvae in growing pigs resulted in high amino acid digestibility, although there was an absence of detailed chitin content.

### 4.2. Benefits of Insect-Derived Chitin in Animal Nutrition

Despite the lower digestibility, the insect chitin content could also have a positive effect on livestock animal health. In general, fish and poultry, which possess a gastrointestinal tract genetically adapted to consume arthropods, typically produce chitinase, suggesting an adaptive mechanism that promotes beneficial effects.

#### 4.2.1. Prebiotic Effect

The most studied benefit of chitin in animal nutrition is its prebiotic effect, the stimulation of the growth and activity of beneficial bacteria in the gastrointestinal tract. Studies on various animals confirmed that insect chitin promotes a shift in intestinal microbiota, enhancing chitin digestion and overall animal health [17,18]. Specifically, the fermentation of chitin encourages the growth of bacteria such as Firmicutes, Clostridiales, Tenericutes, Actinomycetes, chitinolytic bacteria which contribute to the chitin digestion, and lactic acid bacteria (LAB) like *Bifidobacterium* and *Lactobacillus* known for their probiotic activity [13,17,18,55,110]. This microbial modulation can lead to an increase in short-chain fatty acids (SCFAs) such as acetate, propionate, and butyrate, the major products of bacterial fermentation of non-digestible carbohydrates with antimicrobial and anti-inflammatory properties. SCFAs stimulate the proliferation of intestinal mucosal cells, influence metabolism, and play an important role in intestinal physiology [111,112]. Additionally, SCFAs enter the bloodstream and help other organs distant from the digestive system, thereby contributing to overall metabolic balance and gut health of animals [21]. LAB also contribute to the fatty acid composition of meat products enhancing the meat quality and health of consumers. In the context of aquafeed, *H. illucens*, *T. molitor*, and *G. sigillatus* represent the sole edible insect species that have been investigated regarding their influence on intestinal microbial communities (Table 4). In particular, across a wide range of fish species, *H. illucens* inclusion levels from 5% to 45% have consistently increased microbial richness and diversity, with a selective enrichment of LAB-, chitinolytic-, and SCFAs-producing taxa. In rainbow trout, *H. illucens* at 10 and 30% inclusion levels (0.5 and 1.5% chitin) promoted the abundance of Lactobacillaceae, Bacillaceae, Clostridiaceae, and Actinomycetaceae [113,114,115,116,117]. Similar microbial shifts were observed in Siberian sturgeon, Atlantic salmon, and European sea bass with a range of *H. illucens* inclusion levels from 10 to 25% [118,119,120,121]. Rawski et al. [122] further confirmed a dose-dependent increase in *Bacillus*, *Enterococcus*, and *Lactobacillus* across inclusion levels from 5 to 20% in the Atlantic salmon. Similarly, *T. molitor* meal showed clear prebiotic effects with increased level of *Clostridium*, *Lactobacillus*, Lactobacillales, and *Bacillus* reported in rainbow trout, juvenile large yellow croakers, European sea bass, and Siberian sturgeon with moderate and high inclusion levels (from 20 to 100%) [116,118,121,123,124]. However, very high inclusion levels (e.g., 75–100%) may lead to dysbiosis in certain fish species, such as grass carp [125]. Finally, *G. sigillatus* nymphs at 20% inclusion increased LAB and *Clostridium* in rainbow trout [116], suggesting comparable prebiotic potential.

Even in poultry, the inclusion level of insect meals rich in chitin has consistently produced a probiotic effect by changing the gut microbiota to increase beneficial bacteria (Table 5). Research on different types of poultry, such as broiler chickens, slow-growing chickens, ducks, and quails, has found that some edible insect species (*H. illucens*, *T. molitor* and *B. mori*) are linked to higher levels of SCFAs and a healthier gut. For instance, when *H. illucens* was included in low amounts (0.1–0.2%), it led to an increase in *Bacteroides*, *Prevotella*, *Clostridium coccoides*, *Eubacterium*, and LAB such as *Lactococcus* and *Streptococcus* spp., indicating early-stage microbial modulation with a likely prebiotic effect [130]. With a low and moderate inclusion level (5–15%), *H. illucens* continued to stimulate the proliferation of beneficial genera including *Lactobacillus*, *Faecalibacterium*, *Roseburia*, and *Ruminococcus* [131,132,133]. The same effects were observed in slow-growing chickens and Muscovy ducks, demonstrating that *H. illucens*-derived chitin works its efficacy in different types of poultry [134,135]. *Tenebrio molitor* also exhibited strong microbiota-modulating potential. In broilers, both low (0.1–0.3%) and moderate (5–15%) inclusion levels enhanced the abundance of bacterial genera involved in SCFAs production, including *Lactobacillus*, *Enterococcus*, *Alistipes*, *Clostridium*, and Ruminococcaceae [130,132,136,137]. Though less researched, the addition of *B. mori* chrysalides (12.5%) to quail diets stimulated the growth of microbial groups such *Lactobacillus*, *Bacillus*, Eubacteriaceae, and Streptococcaceae that are involved in the fermentation of polysaccharides and the formation of SCFA, indicating a promising prebiotic potential [97].

The prebiotic effect of insect-derived chitin has also been demonstrated in monogastric mammals such as swine. The inclusion of insect meals in pig diets, especially during the weaning and growing phases, has shown clear modulatory effects on the gut microbiota, promoting taxa associated with chitin fermentation, SCFAs production, and intestinal health. In weaned piglets, *H. illucens* at inclusion levels of 4–8% (0.9–1.9% chitin) led to abundance of *Lactobacillus* and several butyrate-producing genera including *Faecalibacterium*, *Clostridium*, *Roseburia*, and *Pseudobutyrivibrio* [138]. Similarly, Biasato et al. [131] reported an increase in populations of *Roseburia*, *Eubacterium*, *Prevotella*, *Blautia*, *Coprococcus*, and *Ruminococcus,* all of which are involved in the degradation of polysaccharides and SCFAs production, in piglets fed 5–10% *H. illucens*. Notably, even a full substitution of fish meal with *H. illucens* significantly enriched beneficial bacterial genera such as *Lactobacillus*, *Faecalibacterium*, *Prevotella*, *Ruminococcus*, and *Alloprevotella*, confirming the prebiotic potential [139]. *T. molitor* also demonstrated similar functional properties in growing pigs; in fact a moderate inclusion level (5–10%) increased the abundance of beneficial genera such as *Bifidobacterium*, *Roseburia*, *Clostridium*, *Faecalibacterium*, *Lactobacillus*, *Fibrobacter*, *Ruminococcus*, *Prevotella*, *Eubaacterium*, and *Blautia* with fibre degradation and SCFAs production activity [140,141].

#### 4.2.2. Immunostimulatory Effect

In addition to modulating the gut microbiota, chitin derived from insects has demonstrated potent immunostimulatory properties. Indeed, it appears to activate the innate immune system of certain fish and poultry species by stimulating pathogen recognition receptors (PRRs), including Toll-like receptors (TLRs). This leads to increased production of antimicrobial peptides and pro-inflammatory cytokines (e.g., IL-1β, TNF-α, IL-6, and IL-8), as well as anti-inflammatory cytokines (e.g., IL-10 and TGF-β), which are useful for preventing infection [13,17,18].

Chitinase may also act as immunostimulant with a potential protective role against bacterial and parasitical infections, also improving the resistance of animals to bacterial diseases. In fact, dietary inclusion levels of insect meal rich in chitin has also been associated with an increase in immunoglobulin levels (e.g., IgM, IgG, and IgA), lysozyme, cytokines, immune-related genes (e.g., MyD88, MHC-II, NF-kB, CypA, HSP70, SIgA), and in albumin-to-globulin ratio, improving immune responses in animals [17]. For that which concerns aquaculture, several studies demonstrated the immunostimulatory effect of insect-derived chitin, particularly through dietary inclusion of *H. illucens* and *T. molitor* (Table 6). Insect meals have been incorporated into a wide range of fish, from rainbow trout, Nile tilapia, koi carp, zebrafish, groupers, catfish, to sturgeon, at levels ranging from low percentages (2.5–5%) to complete dietary replacement (up to 100%) [142,143,144,145]. In most cases, the insect-derived chitin presence improved innate immune responses, evidenced by enhanced lysozyme and myeloperoxidase activity, cytokine up expression (such as TGF-β, IL-10, IL-1β, and TNF-α), and modulation of immune-related genes including IgM, MHC-II, and TLRs [144,145,146,147,148]. In addition, expression of heat shock proteins (e.g., hsp70) and indicators of hepatic health such as reduced aspartate aminotransferase activity have also been reported [142,149]. Increases in white blood cell counts, lymphocytes, and serum proteins including albumin and globulins have typically accompanied these immunomodulatory effects, indicating improved systemic immunity [146,150,151]. Similarly, *T. molitor* meal stimulates immune responses, demonstrating immunostimulatory effects not only with the increase in lysozyme activity and both pro- and anti-inflammatory cytokines, in trout, bass, carp, and catfish [80,125,143,151,152], but also modulating enzymatic responses such as trypsin inhibitors activity in European sea bass, potentially enhancing pathogen clearance [150]. There are not many studies on other species of edible insects. Only Fawole et al. [153] and Bagheri et al. [154] reported an immune-stimulating effect in species like hybrid catfish and beluga sturgeon with a moderate level (from 5 to 20%) of *M. domestica* larvae and *B. mori* chrysalides meal inclusion.

Similarly, the immunostimulant effect of the insect-derived chitin has also been observed in poultry. Among the most studied insects, *H. illucens* demonstrated consistent immunostimulatory properties when incorporated into poultry diets. In broilers, even low inclusion levels (1–3%) led to an increase in lysozyme activity and the proportion of CD3^+^ CD4^+^ T lymphocytes, indicating enhanced cellular immune responses [155]. These findings were also confirmed at a higher inclusion level (75%), which resulted in an increase in CD3^+^ CD4^+^ T-cell counts in blood samples [156]. A comparable immune enhancement has been documented in laying hens, wherein inclusion levels ranging from 3% to 12% promoted a general activation of cellular immunity [157]. For that which concerns *T. molitor*, a complete replacement (100%) of conventional protein sources with *T. molitor* meal increased the albumin-to-globulin ratio in broiler chickens, suggesting a potential improvement in systemic immune function [158]. The same immune stimulation is supported by low inclusion levels (from 0.2 to 2%), with an upregulation of pro-inflammatory cytokines IL-2 and TNF-α and an increase in myeloperoxidase levels [91,159].

In swine nutrition, chitin from *H. illucens* has also demonstrated immunostimulatory properties at varied dietary inclusion levels (1–8%). Pigs fed with *H. illucens* meal showed an increase in TLR-4 and pro-inflammatory cytokines, with higher levels of anti-inflammatory markers such as IL-10. Furthermore, higher serum concentrations of globulin, IgA, and SIgA were observed [104,138,160].

#### 4.2.3. Cholesterol and Triglycerides Reduction

The insect-derived chitin was also responsible for the decrease in serum cholesterol and triglycerides in livestock animals [18,161,162]. According to Khoushab and Yamabhai [163], cholesterol reduction is related to different mechanisms such as the electrostatic interaction between lipids and aminopolysaccharides. In fact, chitin and certain chitin derivatives attach to lipid micelles, blocking their absorption and lowering cholesterol. They also join with bile acids in the digestive tract and excrete them with faeces, increasing bile acid excretion. According to recent research, adding insect meals high in chitin to animal diets consistently reduces serum cholesterol and triglyceride levels. Among species investigated, even in this case, *H. illucens* is the most extensively studied. In striped catfish and European sea bass, high inclusion levels (45 and 60%) led to significant reductions (about 17% and 9%, respectively) in both cholesterol and triglycerides [69,164]. Similar trends were also noted at low inclusion levels. In fact, in Mozambique tilapia a minimal inclusion of 0.2% of *H. illucens* resulted in lower cholesterol levels [165], and in juvenile Jian carp, *H. illucens* meal reduced by up to 10.6% lipid content in the hepatopancreas and serum cholesterol (about 31.5% of reduction) [166]. However, the decrease in both total cholesterol and triglycerides was also observed with a complete inclusion level (100%) in juvenile striped catfish diet [167], demonstrating that *H. illucens* chitin may have a hypocholesterolaemic effect. Other species such as *T. molitor* and *M. domestica* larvae were tested in European sea bass, mandarin fish, and turbot in fish showing a medium–high (10 to 60%) reduction in cholesterol levels [168,169,170]. Similarly, Xu et al. [171] discovered that a complete inclusion level of *B. mori* chrysalides meal (100%) reduced the total cholesterol in juvenile mirror carp by 4.44%. However, this effect was also confirmed in poultry; in fact, *H. illucens* meal induced consistent reductions in serum cholesterol and triglycerides across various studies. For example, Bovera et al. [172] and Montalbán et al. [173] reported a reduction (13.6 and 19.5%) in cholesterol and triglycerides in laying hens, at moderate and high inclusion levels (5 and 50%). In broilers, positive effects were observed starting from 2.5%, with consistent reductions at all tested levels up to 7.5% [174]. Kierończyk et al. [175] reported a significant reduction (about 13.5%) in blood cholesterol only at full replacement (100%), while in broiler ducks a decrease of 32% of high-density lipoprotein (HDL), 39% of low-density lipoprotein (LDL), 47% of cholesterol was observed with an inclusion of 7.5% [176]. The effect of *T. molitor* was also notable in poultry; Nassar et al. [157] reported that in broilers inclusion levels up to 6% were effective in reducing 4.44 and 3.35% of both LDL cholesterol and triglycerides, and in laying hens, there was a 5% inclusion reduced total cholesterol content [177]. As for swine, the available data are still very limited. Only one study on growing pigs reported an increase of 22.6% in serum cholesterol with a 10% inclusion of *T. molitor* meal compared to the control group [178].

#### 4.2.4. Antimicrobial Effect

Finally, insect-derived chitin is also recognised for its antimicrobial effect against bacteria, fungi, and yeast in poultry and pigs [18]. In particular, chitin seems to decrease in mortality and reduce the levels of various pathogens (especially bacteria) acting as a natural antibiotic and increase animal health [17]. For that which concerns fish, no studies were directly reported on the antimicrobial effect of insect-derived chitin [17]. As far as the authors can ascertain, Abd El- Gawad et al. [146] is the only study which reports that the use of *H. illucens* as 100% inclusion in Nile tilapia meal effectively protects the health of fish when exposed to *Streptococcus iniae* infection, enhancing haemato-immunological parameters, antioxidant capability, and anti-inflammatory gene expression. Other studies reported an improvement of lysozyme and chitinase activity when barramundi and Nile tilapia, infected with *Vibrio harveyi* and *Escherichia coli*, respectively, were fed with different inclusion level of *H. illucens* (from 10, 15 and 60%), indicating a possible reduction in pathogens [179,180]. A similar scenario was also observed for poultry. In fact, Lee et al. [155] reported an enhancement of immune activities and survivability of broiler against *Salmonella gallinarum* infection when fed with 1, 2 and 3% inclusion level of *H. illucens* meal in the diet. In that case, the authors also observed a decrease in the Gram-negative bacteria in liver, spleen and caeca. The same effect was observed by Isam and Yang [181] who reported a decrease in mortality, increase in immunoglobulin level and reduction in *E. coli* and *Salmonella enteritis* colonies when broilers were fed with 0.4% of *T. molitor*, assuming that it can be used as an alternative antibiotic in broiler diets. Spranghers et al. [182] reported that low and moderate inclusion levels of *H. illucens* (4, 5 and 8%) reduce the infection of *D-streptococci* in the gut of piglets, demonstrating that future research should focus on exploring potential of insect meal as antibiotics. However, current studies are not sufficient, and the mechanism of insect-derived chitin meal against pathogens is not yet fully clear. Khoushab and Yamabhai [163] indicate that the antimicrobial activity of insect-derived chitin seems to be related to the accumulation of chitinases in the animal intestine, or to the disruption of bacterial cell walls caused by interactions between chemical charges.

## 5. Conclusions

Growing interest in edible insects as sustainable animal nutrition ingredients has brought attention to less-studied components, such as chitin, in addition to boost their protein and fat levels. According to this review, chitin derived from insects has a dual and complex role in animal nutrition. It can be an antinutritional factor because of its poor digestibility, but it can also be a functional compound that has antimicrobial, immunostimulatory, cholesterol-lowering, and prebiotic properties for fish, poultry, and pigs. Chitin digestibility remains one of the main limiting factors for the inclusion of insect meals in animal diets. However, beyond digestibility, chitin is increasingly recognised for its functional effects on animal health. Chitin-rich insect meals have shown the ability to modulate gut microbiota, promoting the increase in short-chain fatty acids (SCFAs) producing bacteria and lactic acid bacteria (LAB). In addition to gut health, several studies have highlighted the immunostimulatory potential of chitin. The inclusion of insect meal rich in chitin has been associated with increased expression of cytokines (e.g., IL-1β, TNF-α, IL-10), enhancement of lysozyme and immunoglobulin levels, and upregulation of immune-related genes such as TLR-4, MyD88, and MHC-II, particularly in fish and poultry species. These findings reinforce the idea that chitin may act not only as a dietary fibre, as well as an active immunomodulator for animal health. Furthermore, it has been demonstrated that chitin derived from insects can lower triglycerides and cholesterol levels in fish and poultry. Additionally, several studies also highlight the antimicrobial potential of chitin with evidence of reducing pathogens and increasing survivability in animals exposed to bacterial infection.

However, despite the encouraging evidence, some knowledge gaps remain. Most studies focus on a limited number of edible insect species, such as *H. illucens* and *T. molitor*, while the effects of chitin from other authorised insect species, *A. diaperinus*, *A. domesticus*, *G. assimilis*, *G. sigillatus*, *M. domestica*, and *B. mori*, are still underexplored. Furthermore, most studies focus on growth or digestibility outcomes; long-term impacts on animal health, microbiota, and production quality remain to be addressed. Future research should aim to clarify dose–response thresholds for both beneficial and adverse effects of chitin across different animal species and expand studies on the chitin effect from the other EU-authorised insect species on fish, poultry and swine nutrition.

In conclusion, chitin from edible insects should no longer be viewed only as a structural component, but as an active ingredient that can benefit animal nutrition and health in many ways. Finding a balance between its nutritional value and its functional impacts is crucial to maximising its potential and assisting in the development of healthier and more sustainable animal diets.

## Figures and Tables

**Figure 1 insects-16-00799-f001:**
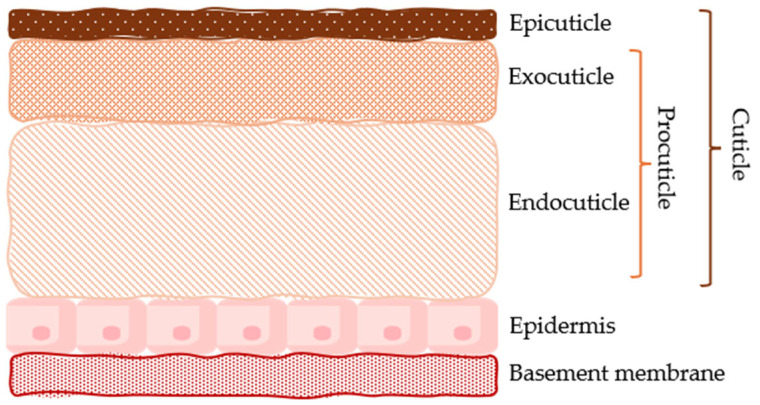
The main layers of the insect exoskeleton.

**Figure 2 insects-16-00799-f002:**
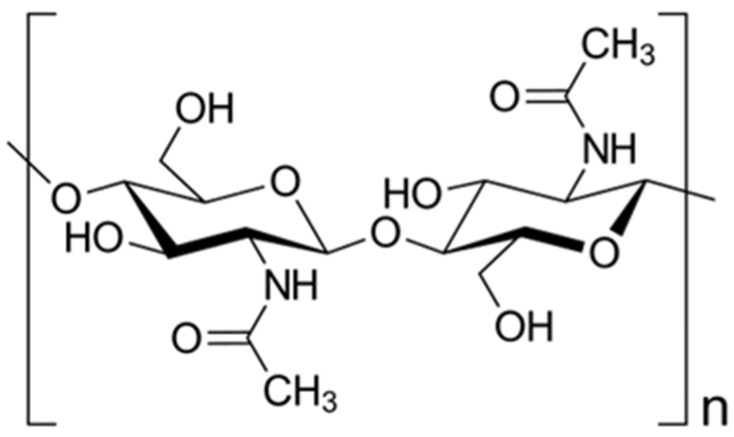
Chitin structure.

**Figure 3 insects-16-00799-f003:**
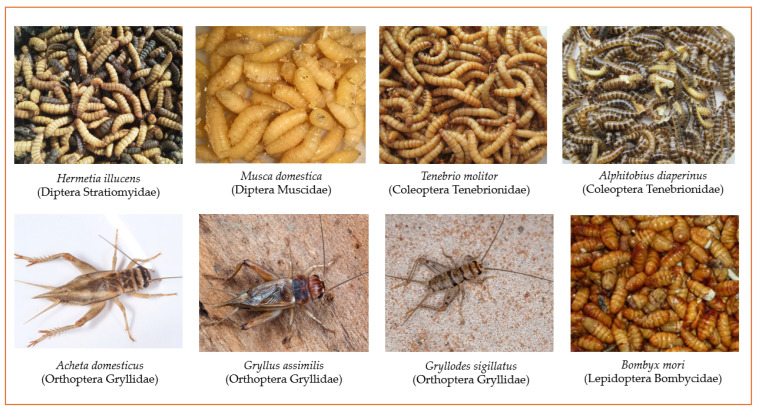
Edible insect species authorised in European Union for feed production. (Picture by Dave Rentz: *Acheta domesticus*; Steve Covey: *Gryllus assimilis*; ViniSouza128: *Gryllodes sigillatus*).

**Table 1 insects-16-00799-t001:** Chitin content in the edible insect species allowed for animal nutrition in the European Union.

Order	Species	Instars	Chitin Content (%)	References
Coleoptera	*Tenebrio molitor* (Tenebrionidae)	Larvae	4.60	Shin et al. [46]
		Pupae	3.90	
		Adults	8.40	
	*Alphitobius diaperinus* (Tenebrionidae)	Larvae	4.2–6.2	Soetemans et al. [47]
Diptera	*Hermetia illucens* (Stratiomyidae)	Larvae	13	Smets et al. [48]; Triunfo et al. [49]
		Prepupae	4.7	
		Exuviae	31	
		Adults	9	
	*Musca domestica* (Muscidae)	Larvae	9.1	Zhang et al. [50] Kim et al. [51]
		Pupae	8.02	
Lepidoptera	*Bombyx mori* (Bombycidae)	Chrysalides	15–20	Zhang et al. [43]
Orthoptera	*Acheta domesticus* (Gryllidae)	Adults	4.3–7	Ibitoye et al. [52]
	*Gryllodes sigillatus* (Gryllidae)	Adults	3.4	Malm [53]
	*Gryllus assimilis* (Gryllidae)	Adults	2.9–3.3	Toribio et al. [54]

**Table 2 insects-16-00799-t002:** Summary of studies on the effect of chitin, derived from edible insect species authorised in European Union for feed, on the apparent digestibility coefficients of nutrients by different fish species.

Insect Species	Instars	Fish Species	Inclusion Level (%)	Chitin Content (%)	Effect on Digestibility	References
*Hermetia * *illucens*	Larvae	Rainbow trout	25, 50%	1, 2%	Dose-dependent low ADC of CP, dry matter and PUFAs	Renna et al. [57]
	Larvae		6.6, 13.2, 26.4%	/	No significant differences in the ADC of CP and essential AAs. Dose-dependent low digestibility of lipids, dry matter, and taurine	Dumas et al. [75]
	Larvae		/	8%	No significant differences in the ADC of CP, EE, and chitin	Ushakova et al. [76]
	Larvae		25% with three different types of fractions	1.8, 2.7, 15.4%	Dose-dependent low ADC of dry matter, CP and nitrogen-free extract	Eggink et al. [62]
	Larvae		15%	/	No significant differences in the ADC of CP, nitrogen, lipids, and energy, with high levels of lauric and myristic fatty acids	Drosdowech et al. [71]
	Larvae	Nile tilapia	25% with three different types of fractions	1.8, 2.7, 15.4%	Dose-dependent low ADC of dry matter, CP and nitrogen-free extract	Eggink et al. [62]
	Larvae	Red hybrid tilapia	30%	1%	High ADC of proteins, and GE	Muin and Taufek [77]
	Larvae	Red tilapia	/	7.8%	No significant differences in the ADC of CP, EE, and chitin	Ushakova et al. [76]
	Larvae	Meagre	17, 35, 52%	0.6, 1.1, 1.6%	Dose-dependent low ADC of dry matter, CP, and some essential and non-essential AAs	Guerreiro et al. [61]
	Larvae	African catfish hybrid juveniles	30%	9%	High ADC of CP, EE, ash and phosphorus, less digestibility of essential AAs	Sándor et al. [63]
	Prepupae	Juvenile red sea bream	15, 30 and 45%	/	No significant differences in the ADC of nutrients except for lipids during complete replacement	Oktay et al. [70]
	Larvae	Juvenile turbot	17, 33, 49, 64 and 76%	From 1.6 to 7.3%	Dose-dependent low ADC of CP and GE	Kroeckel et al. [65]
	Larvae	Russian sturgeon	/	7.8%	No significant differences in the ADC of CP, EE, and chitin	Ushakova et al. [76]
	Prepupae	European seabass	15, 30, 45%	/	No significant differences in the ADC of CP, EE	Magalhães et al. [69]
	Larvae	Atlantic salmon	33, 66, 100%	/	No significant differences in the ADC of CP, EE, AAs and fatty acids, or the digestive enzyme	Belghit et al. [68]
*Tenebrio molitor*	Larvae	Rainbow trout	25, 50%	/	Dose-dependent low ADC of CP	Belforti et al. [56]
	Larvae		30%	/	No differences in the ADC of CP, lipids, dry matter, and GE	Owens et al. [78]
	Larvae	Nile tilapia	21, 43%	1.37, 2.82%	No differences in the ADC of CP	Sánchez-Muros et al. [58]
	Larvae		20%	3.8%	High ADC of CP, dry matter, and chitin	Fontes et al. [60]
	Larvae	Meagre	10, 20, 30%	0, 74, 0, 97, 1.47%	Dose-dependent low ADC of dry matter, GE, CP, and AAs	Coutinho et al. [66]
	Larvae	African catfish hybrids	30%	6%	Low ADC of dry matter, AAs, fatty acids	Sándor et al. [63]
	Larvae	European sea bass	25, 50%	/	No differences in the ADC of dry matter, CP, and EE	Gasco et al. [79]
	Larvae	Gilthead sea bream	25, 50%	1.15, 2.31%	Dose-dependent low ADC of CP and EE	Piccolo et al. [59]
	Larvae	Juvenile largemouth bass	12, 24, 36, 48%	/	High ADC of CP, lipids, and dry matter	Chen et al. [80]
*Acheta* *domesticus*	Adults	Rainbow trout	30%	/	No differences in the ADC of CP, lipids, dry matter, and GE	Owens et al. [78]
*Alphitobius* *diaperinus*	Larvae	Rainbow trout	30%	7%	Low ADC of CP, AAs, dry matter, and GE	Gasco et al. [64]
*Gryllus* *assimilis*	Adults	Nile tilapia	20%	5%	Low ADC of dry matter, CP, and GE	Fontes et al. [60]
*Gryllodes* *sigillatus*	Adults	Rainbow trout	15%	/	No significant differences in the ADC of CP, nitrogen, lipid, and GE	Drosdowech et al. [71]
*Musca* *domestica*	Larvae	Nile tilapia	9, 18, 27, 36%	/	No significant differences in the ADC of dry matter, CP, lipids, GE, and phosphorus	Wang et al. [72]
*Bombyx* *mori*	Chrysalides	Common carp	10, 20, 30%	/	No significant differences in the ADC of CP and EE	Nandeesha et al. [73]
	Chrysalides	Juvenile mirror carp	4, 8, 12, 16%	/	No significant differences in the ADC of dry matter, CP, and EE	Zhou et al. [74]

**Table 3 insects-16-00799-t003:** Summary of studies on the effect of chitin, derived from edible insect species authorised in European Union for feed, on the apparent digestibility coefficients of nutrients by poultry.

Insect Species	Instars	Poultry	Inclusion Level (%)	Chitin Content (%)	Effect on Digestibility	References
*Hermetia illucens*	Larvae	Broilers	25%	/	High ADC of dry matter, CP, and AAs	De Marco et al. [88]
	Larvae		2, 4, 6, 8, 10%	/	No significant differences in the ADC of EE and dry matter, high ADC of CP	Kareem et al. [90]
	Prepupae		5%	/	No significant differences in the ADC of CP	Elangovan et al. [93]
	Larvae		100%	5.5%	Low ADC of fibres and EE	Chobanova et al. [86]
	Larvae	Quails	10, 15%	/	No significant difference in the ADC of dry matter, CP, and GE, except for EE which was less for 10% inclusion level	Cullere et al. [89]
	Larvae	Layers	3, 6, 9%	/	No significant differences in the ADC of dry matter, CP, EE, calcium, and phosphorous	Chu et al. [92]
	Larvae	Laying hens	15%	7%	Low ADC of AAs	Heuel et al. [85]
	Larvae	Sentul chickens	2, 4, 6%	/	Dose-dependent high ADC of CP and EE	Rahayu et al. [95]
*Tenebrio molitor*	Larvae	Broilers	25%	/	High ADC of dry matter, CP, and AAs	De Marco et al. [88]
	Larvae		100%	4.62%	Low ADC of dry matter and CP	Bovera et al. [83]
	Larvae		0.2, 0.3%	8.9%	No significant differences in the ADC of CP and EE	Benzertiha et al. [91]
	Larvae		20%	/	Low ADC of dry matter and CP, except for EE and GE	Dourado et al. [84]
	Larvae		30%	/	High ADC of dry matter, GE, and AAs	Nascimento et al. [94]
*Gryllus* *assimilis*	Nymphs	Broilers	20%	/	Low ADC of dry matter and CP, except for EE and GE	Dourado et al. [84]
*Musca* *domestica*	Larvae	Broilers	5, 10, 15, 20%	/	High ADC of CP and AAs	Hwangbo et al. [87]
*Bombyx mori*	Chrysalides	Fattening quails	12.5%	2.8–3.5%	Low ADC of dry matter, CP, EE, and GE	Dalle Zotte et al. [97]
	Chrysalides	Broilers	4%	/	No significant differences in the ADC of dry matter and CP	Singh et al. [98]

**Table 4 insects-16-00799-t004:** Prebiotic effect of chitin from edible insect species authorised in European Union for aquafeed.

Insect Species	Instars	Fish Species	Inclusion Level (%)	Chitin Content (%)	Prebiotic Effect	References
*Hermetia illucens*	Larvae	Rainbow trout	30%	/	Increase in richness of microbiota with LAB such as Bacillaceae	Huyben et al. [115]
	Prepupae		10, 20, 30%	0.50, 0.99, 1.51%	Enhancement of diversity and abundance of gut microbiota, with increased mycoplasma linked to LAB production capacity	Rimoldi et al. [117]
	Prepupae		10, 20, 30%	0.5, 0.9, 1.5%	Increase in richness and diversity of microbiota and increase in LAB and butyrate production bacteria such as Lactobacillaceae, Bacillaceae, Actinomycetaceae, and Clostridiaceae	Terova et al. [114]
	Larvae		20%	/	Increase in richness of microbiota with *Clostridium* and LAB	Józefiak et al. [116]
	Exuviae		1.6%	/	Increase in richness of gut microbiota with *Bacillus*, *Facklamia*, *Brevibacterium*, and *Corynebacterium* genera with chitinolytic and lactic acid production activity	Rimoldi et al. [113]
	Larvae		15%	/	Increase in richness and diversity of microbiota and increase in *Peptostreptococcus* with prebiotic effect and digestion and fermentation activity	Drosdowech et al. [126]
	Larvae	Siberian sturgeon	15%	/	Increase in richness of microbiota with *Bacillus*, *Enterococcus*, * Lactobacillus*, and Entrerobacteriaceae	Józefiak et al. [118]
	Larvae	Atlantic salmon	10%	/	Increase in LAB and chitin degrading bacteria of genus *Exoguobacterium*	Leeper et al. [119]
	Larvae	Atlantic salmon	20%	1.44%	Increase in LAB such as Actinomycetaceae, Lactobacillaceae, and chitinolytic bacteria such as Bacillaceae and Actinomycetaceae	Weththasinghe et al. [120]
	Larvae	Atlantic salmon	5, 10, 15, 20%	/	Increase in *Bacillus*, *Enterococcus* and *Lactobacillus* with chitinolytic and prebiotic effect	Rawski et al. [122]
	Larvae	European sea bass	25%	1.8%	Increase in Firmicutes, Bacillaceae, Enterococcaceae, Lachnospiraceae, and Actinomycetaceae with chitinolytic activity and prebiotic effect	Rangel et al. [121]
	Larvae	Gilthead sea bram	5, 10, 15%	/	Increase and shift in microbiota with abundance of Bacillaceae and Paenibacillaceae involved in chitin degradation and prebiotic effect	Busti et al. [127]
	Larvae	Gilthead seabream juveniles	15, 30, 45%	/	Increase in richness and diversity of gut microbiota promoting beneficial digesta bacteria	Moutinho et al. [128]
	Larvae	Hybrid grouper	10, 30, 50%	/	Increase in gut microbiota diversity with *Thiobacillus*, *Sutterella*, *Veillonella*, *Dialister* and *Biophila* genera, and Hydrogenophilaceae family associated with beneficial metabolic functions	Chen et al. [129]
*Tenebrio molitor*	Larvae	Rainbow trout	20%	/	Increase in richness of microbiota with *Clostridium* and LAB	Józefiak et al. [116]
	Larvae		100%	1.49%	Increase in richness and diversity of microbiota with abundance of Lactobacillales	Terova et al. [123]
		Juvenile large yellow croakers	15, 30, 45%	/	Increase in relative abundance of *Bacillus* and *Lactobacillus*	Zhang et al. [124]
	Larvae	Grass carp	25, 50, 75, 100%	/	25% inclusion positively affected beneficial intestinal bacteria, while higher levels disrupted gut microbiota increasing harmful bacteria like *Brevinema*	Yang et al. [125]
	Larvae	Siberian sturgeon	15%	/	Increase in richness of microbiota with *Bacillus*, *Enterococcus*, and *Lactobacillus*	Józefiak et al. [118]
	Larvae	European sea bass	25%	/	Increase in the relative abundance of beneficial and chitinolytic bacteria	Rangel et al. [121]
*Gryllodes sigillatus*	Nymphs	Rainbow trout	20%	/	Increase in richness of microbiota with *Clostridium* and LAB	Józefiak et al. [116]

**Table 5 insects-16-00799-t005:** Prebiotic effect of chitin from edible insect species authorised in European Union in poultry.

Insect Species	Instars	Poultry	Inclusion Level (%)	Chitin Content (%)	Prebiotic Effect	References
*Hermetia* *illucens*	Larvae	Broiler	0.1, 0.2%	/	Increase in *Bacteroides*, *Prevotella*, *Clostridium* coccoides, *Eubacterium*, *Streptococcus* spp. and *Lactococcus* spp., with prebiotic effect and beneficial effect	Józefiak et al. [130]
	Larvae		5, 10, 15%	/	Moderate inclusion level increases beneficial microbiota such as L-*Ruminococcus*, *Lactobacillus*, *Faecalibacterium*, *Blautia*, *Roseburia,* and *Clostridium* with lactic acid activity and SCFA production	Biasato et al. [131]
	Larvae		5%	/	Increase in *Victivillaceae*, *Saccharibacteri*, *Clostridium*, and *Eubacterium* involved in polysaccharide fermentation and SCFA production	Colombino et al. [132]
	Larvae		5, 10, 15, 20%	/	Abundance of the bacterial group *Roseburia*, known for the SCFA production	De Souza Vilela et al. [133]
	Larvae	Slow-growing chickens	5%	/	Increase in beneficial bacteria, such as *Faecalibacterium*, known to produce SCFA	Fiorilla et al. [134]
	Larvae	Muscovy ducks	3, 6, 9%	/	Increase in *Faecalibacterium*, *Megamonas*, and *Ruminococcus*, known for the SCFA production and beneficial effect	Martínez Marín et al. [135]
*Tenebrio molitor*	Larvae	Broiler	5, 10, 15%	/	Increase in abundance of *Clostridium*, *Alistipes*, and *Sutterella* with beneficial effect and butyric acid production	Biasato et al. [136]
	Larvae		0.1, 0.2%	/	Increase in *Lactobacillus* spp., *Enterococcus*	Józefiak et al. [130]
	Larvae		0.2, 0.3%	/	Increase in abundance of Ruminococcaceae and *Lactobacillus*	Józefiak et al. [137]
	Larvae		5%	/	Increase in *Collinsella* and *Eubacterium* with beneficial effect and SCFA production	Colombino et al. [132]
*Bombyx mori*	Chrysalides	Fattening quails	12.5%	1–1.40%	Increase in Streptococcaceae, Rikenellaceae, Eubacteriaceae, *Lactobacillus*, and *bacillus* involved in polysaccharide fermentation, SCFA production, and prebiotic effect	Dalle Zotte et al. [97]

**Table 6 insects-16-00799-t006:** Immunostimulatory effect of chitin from edible insect species European Union authorised in aquaculture.

Insect Species	Instars	Fish Species	Inclusion Level (%)	Immunostimulatory Effect	References
*Hermetia* *illucens*	Larvae	Rainbow trout	25, 50%	Inhibitory activity of pathogen bacterial, aspartate blood aminotransferase lower and lysozyme content higher compared to the control	Hwang et al. [142]
	Larvae		25, 50, 75, 100%	Increase in the expression of cytokines (TGF, IL-10, IL-1β, TNF−α, and IL-8) and immune-related genes (IgM, IgT, MHC-II, and TRL-5) in all insect meals compared to the control	Sayramoğlu et al. [144]
	Larvae	Nile tilapia	33, 100%	Increase in lysozyme activity, white blood cell count, lymphocyte count at 100% inclusion level compared to the control. Even globulin and albumin increased in both treated groups	Abd El-Gawad et al. [146]
	Larvae	Koi carp	5, 10, 15, 20%	Increase in mRNA transcripts of immune-related genes such as TNF-α, TGF-β, IL1, IL10, and hsp70	Linh et al. [149]
	Larvae	Zebrafish	2,5, 3, 10%	Increase in TNF-α, IL-10, and hsp70 genes	Zhang et al. [145]
	Larvae	Pearl gentian grouper	10, 20, 30%	Increase in immune response as shown by the increase in key genes NF-κB, TLR, MyD88, IL-10	Huang et al. [147]
	Larvae	Juvenile grouper	2.5, 5, 10%	Increase in lysozyme activity, TNF-α, hsp70, and IL-1β genes	Jian et al. [148]
*Tenebrio molitor*	Larvae	Rainbow trout	7, 14, 21, 28%	Increase in lysozyme activity and myeloperoxidase	Jeong et al. [143]
	Larvae	European sea bass	36%	Increase in lysozyme antibacterial activity with inhibition of serum trypsin inhibition	Henry et al. [150]
	Larvae	Juvenile largemouth Bass	12, 24, 36, 48%	Increase in anti-inflammatory genes such as IL-10 and TGF and pro-inflammatory genes such as IL-8 and IL-1β	Chen et al. [80]
	Larvae	Grass carp	25, 50, 75, 100%	Increase in cytokines with anti-inflammatory responses such as NF-kB, IL- β, IL-6 and TNF-α	Yang et al. [125]
	Larvae	Large yellow croaker	15, 30, 45, 60, 75, 100%	Increase in Keap-1, NF-kB, and TNF-α in intestine and liver	Qu et al. [152]
	Larvae	Yellow catfish	25, 50, 75%	Increase in immune-related genes as MHC II, IL-1, CypA, IgM, HEq	Su et al. [151]
*Musca* *domestica*	Larvae	Hybrid catfish	14, 21%	Improving immune physiology with white blood cell, lymphocyte count and globulin increase	Fawole et al. [153]
*Bombyx mori*	Chrysalides	Beluga sturgeon	5, 10, 15%	Increase in lysozyme and IgM activity	Bagheri et al. [154]

## Data Availability

Data are available on request from the corresponding authors.

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
