# Peer review of "Harnessing Chitin from Edible Insects for Livestock Nutrition"

_insects, 2025, doi:10.3390/insects16080799_

Round 1
Reviewer 1 Report
Comments and Suggestions for Authors
The review summarizes the current knowledge on the role of chitin derived from insects in animal nutrition. As it is known this component exhibits both negative and positive effects and the exploration of both is necessary in order to properly manage the nutrient strategies including insects.
The abstract of the Review allows independent reading. The introduction provides the necessary background justifying the aim of the work. A small remark:
Line 41 -42- please, correct: Asia, Africa and Latin America are not countries. So, please, rephrase the sentence.
The aim of the review is very clearly defined.
The review focuses on the structure of chitin and its content is several most widely used insect species. The section is well illustrated with figures and tables.
Further the authors describe in details the negative and positive effects of chitin in the nutrition of monogastric animals, poultry and fish. The review includes relevant studies and provides a well structured summary of the problem. The future perspectives are very well outlines in the conclusion section.
The latter , however is rather vast and might be shortened.
Author Response
Thank you to the reviewer for their comments and suggestions. Regarding the remark on lines 41–42, we fully agree with the observation.
The phrase has been revised; the sentence now reads:
“In parts of the world such as Asia, Africa, and Latin America, where entomophagy is a traditional practice, insects are often consumed whole and prepared using various techniques such as frying, boiling, and drying.”
We also have carefully shortened the conclusion section, as requested.
We appreciate the attention to detail and constructive feedback, which helped us to improve the clarity and accuracy of the manuscript.
Reviewer 2 Report
Comments and Suggestions for Authors
I want to thank authors for their efforts in writing this review. the review was well organized and comprehensively described. However minor points needs to be revised.
1- line 12: remove on the other hand
2- line 49 -51: the reference needs to be more recent like (Khalifah, A.; Abdalla, S.; Rageb, M.; Maruccio, L.; Ciani, F.; El-Sabrout, K. Could Insect Products Provide a Safe and Sustainable Feed Alternative for the Poultry Industry? A Comprehensive Review. Animals 2023, 13, 1534. https://doi.org/10.3390/ani13091534 ) instead of ref no 5 and 6
3- Line71: I think in this point the author may add a fig. to describe the layers of insect exoskeleton as this will improve the review.
4- line 168: remove the word on the other hand
5- (tables 3,4 and 5) i think tables are too long. i prefer to summarize tables and add only the references that contain the chitin percentage content (or keep 2 or 3 without chitin percentage content)
6- (table 6) if the chitin percentage is known please add to the table
7- line 473: conclusion is too long please summarized it with the main points only
8- line 528: please rearrange the abbreviations alphabetically.
Author Response
We sincerely thank the reviewer for the positive evaluation of the manuscript and the suggestions. Please find below our responses to the comments:
1. Line 12: remove on the other hand.
R: We have removed the expression "on the other hand" from line 12, as requested.
2. Line 49 -51: the reference needs to be more recent, like (Khalifah, A.; Abdalla, S.; Rageb, M.; Maruccio, L.; Ciani, F.; El-Sabrout, K. Could Insect Products Provide a Safe and Sustainable Feed Alternative for the Poultry Industry? A Comprehensive Review. Animals 2023, 13, 1534. https://doi.org/10.3390/ani13091534 ) instead of ref no 5 and 6.
R: Thank you for the suggestion. We have replaced reference 6 with the reference suggested.
3. Line 71: I think in this point the author may add a fig. to describe the layers of insect exoskeleton as this will improve the review.
R: A figure (Figure 1) illustrating the structural layers of the insect exoskeleton has been added (line 76), as suggested, to enhance the clarification of the main layers of the insect exoskeleton.
4. Line 168: remove the word on the other hand.
R: We removed the phrase "on the other hand" from line 168 to improve the sentence clarity.
5. (tables 3,4 and 5) i think tables are too long. i prefer to summarize tables and add only the references that contain the chitin percentage content (or keep 2 or 3 without chitin percentage content).
R: We thank the reviewer for the valuable suggestion. However, we have decided to keep tables 3, 4, and 5 as they are because they contain important information and studies not covered in the main text, which contribute to a more comprehensive review. To improve clarity, we have reduced the content within the tables by introducing abbreviations.
6. (table 6) if the chitin percentage is known please add to the table.
R: Unfortunately, the cited works do not report the chitin percentage. For this reason, we did not include a column for this data in the table, as it would have remained empty.
7. Line 473: conclusion is too long please summarized it with the main points only.
R: We thank the reviewer for the valuable suggestion. We have carefully revised and shortened the conclusion section, focusing on the main points as requested.
8. Line 528: please rearrange the abbreviations alphabetically.
R: All abbreviations have been rearranged in alphabetical order as requested.
We are grateful for your constructive feedback and are confident that the revisions have enhanced the manuscript’s overall quality.